# Modeling the Carbon Sequestration Potential of Multifunctional Agroforestry-Based Phytoremediation (MAP) Systems in Chinandega, Nicaragua

Elisie Kåresdotter [1,*], Lisa Bergqvist [2,3], Ginnette Flores-Carmenate [4], Henrik Haller [2] and Anders Jonsson [2]

1   Department of Physical Geography and Bolin Centre for Climate Research, Stockholm University, 106 91 Stockholm, Sweden
2   Department of Ecotechnology and Sustainable Building Engineering, Mid Sweden University, 831 25 Östersund, Sweden; lisa.bergqvist@su.se (L.B.); henrik.haller@miun.se (H.H.); anders.jonsson@miun.se (A.J.)
3   Stockholm University Baltic Sea Centre, Stockholm University, 106 91 Stockholm, Sweden
4   AB Hjortens Laboratorium, 831 48 Östersund, Sweden; ginnette.flores@hjortenslab.se
*   Correspondence: elisie.karesdotter@natgeo.su.se

**Abstract:** Global sustainability challenges associated with increasing resource demands from a growing population call for resource-efficient land-use strategies that address multiple sustainability issues. Multifunctional agroforestry-based phytoremediation (MAP) is one such strategy that can simultaneously capture carbon, decontaminate soils, and provide diverse incomes for local farmers. Chinandega, Nicaragua, is a densely populated agricultural region with heavily polluted soils. Four different MAP systems scenarios relevant to Chinandega were created and carbon sequestration potentials were calculated using CO2FIX. All scenarios showed the potential to store significantly more carbon than conventional farming practices, ranging from 2.5 to 8.0 Mg $CO_2$eq ha$^{-1}$ yr$^{-1}$. Overall, carbon sequestration in crops is relatively small, but results in increased soil organic carbon (SOC), especially in perennials, and the combination of crops and trees provide higher carbon sequestration rates than monoculture. Changes in SOC are crucial for long-term carbon sequestration, here ranging between 0.4 and 0.9 Mg C ha$^{-1}$ yr$^{-1}$, with the most given in scenario 4, an alley cropping system with pollarded trees with prunings used as green mulch. The adoption rate of multifunctional strategies providing both commodity and non-commodity outputs, such as carbon sequestration, would likely increase if phytoremediation is included. Well-designed MAP systems could help reduce land-use conflicts, provide healthier soil, act as climate change mitigation, and have positive impacts on local health and economies.

**Keywords:** phytoremediation; carbon sequestration; multifunctional land use; nature-based solution; agroforestry; climate change mitigation

## 1. Introduction

About half of the world's cultivable land area is used for agriculture [1] and a further global expansion of the agricultural frontier is not possible without crossing the biophysical thresholds of the planetary boundaries [2,3]. Since the global population is growing and getting wealthier and demands more food, biomass-based energy, construction wood, and other biomaterials, the pressure on the land carrying capacity is increasing [4]. However, the productivity of the agricultural land is threatened by loss of topsoil, soil pollution, floods, drought due to climate change, etc. [5–7]. The complexity of the global sustainability challenges and the diminishing land resources call for energy-efficient strategies that address several sustainability issues simultaneously and supply more functions from each unit of area, i.e., multifunctional land use. Multifunctional land-use systems can avoid land-use conflict by merging economic, social, and environmental foci, and the production includes non-commodity outputs such as resource protection, carbon sequestration,

erosion control, groundwater recharge, remediation of polluted soil as well as commodity outputs such as energy biomass or food [8–11]. In Nicaragua and many other low-income countries in the Global South, authorities often fail to adequately address environmental concerns such as soil pollution [12,13]. Chinandega in western Nicaragua is one of the most important agricultural and densely populated municipalities where large fractions of the soil are heavily polluted by toxaphene and other persistent organic pollutants (POP) from pesticides used in former cotton fields during the cotton boom from the 1950s to the 1990s [14–16]. These pollutants are prone to bioaccumulation in the food chain, which, together with their hydrophobic nature and low rate of degradation, pose potential threats to human health and ecosystem integrity [15,16]. Due to their persistence, high concentrations of pollutants are still found in the soil, making large areas potentially inappropriate for food production. Conventional soil remediation methods tend to be energy- and labor-intensive, and due to the size of the contaminated areas and low interest from authorities or investors, traditional soil remediation technology would be cost-prohibitive in Chinandega. However, less labor- and energy-intensive, in situ technologies such as phytoremediation (i.e., the use of plants and plant-associated soil microbes to reduce concentrations or toxic effects of environmental pollutants) offer opportunities in such areas where lack of resources or incentives limits the materialization of soil remediation projects. In addition to soil pollution, Chinandega also faces multiple sustainability challenges, including biodiversity loss, climate change, and deforestation. Nature-based solutions (NBS), such as multifunctional agroforestry-based phytoremediation (MAP) systems, that can address several of these challenges simultaneously are thus potentially an appropriate way to create enough socioeconomic incentives for remediation projects to materialize in this context. Agroforestry, i.e., the intentional cultivation of trees and crops in interacting combinations, has a documented capacity for carbon sequestration, erosion prevention, soil fertilization, phytoremediation, increased biodiversity, and may provide income security and income diversification for local farmers [17–23]. Furthermore, agroforestry can function as a climate change adaptation measure by enabling a sustainable intensification of farming with higher productivity as water and nutrient efficiency are improved, and soil fertility is maintained, without using a shifting agriculture system to preserve crop yields [24]. Agroforestry has been shown to have significantly higher carbon sequestration potential compared to conventional farming methods in the tropics [21–23]. Alley cropping (cultivation of crops between rows of trees) and shading systems (crop grown under shadowing trees) are two agroforestry systems that have been successfully used in Central America [18,25]. Depending on the species, the trees can be pollarded regularly for soil improvement or left to grow to maturity to produce firewood and timber [26]. Trees in general [27] and fast-growing trees in particular [28] have shown promise for phytoremediation purposes, especially in short-rotation pollarding systems [29]. The objective of this study is to (1) evaluate the potential for carbon sequestration of four different agroforestry-based phytoremediation systems comprised of appropriate phytoremediation species for Chinandega and (2) assess the suitability of the different MAP systems. As agroforestry has high carbon sequestration potential in the tropics, the hypothesis is that the four MAP systems will display a significant carbon sequestration potential.

## 2. Materials and Methods

Four scenarios of MAP systems were created based on species present in Chinandega, whose carbon sequestration potential was assessed by the software CO2FIX. Chinandega has a tropical savannah climate with a pronounced dry season (November–April) during which temperatures as high as 42 °C are reached. The annual precipitation is sometimes as low as 500 mm and most of the rain falls during the wet season (May–October).

### 2.1. The Study Sites

The study sites were chosen because they represent different concentrations of pollution: high (El Picacho) and moderate (El Ensayo), and due to their history of exposure to toxaphene and other pesticides. A number of pesticides are present at both sites including toxaphene, dieldrin, lindane, endosulfane, parathion, and DDT. At El Picacho, concentrations as high as 5600 mg/kg have been detected but at El Ensayo, most pesticides were found in concentrations <3.5 mg/kg soil.

#### 2.1.1. The Picacho Airfield

El Picacho is a small public airfield (an area of <1 ha). The site was used until the late 1980s for spray planes and it is one of the most heavily contaminated sites in Chinandega. The contamination originates from the spray pumps carrying the pesticides that were tested and refilled at the airport before being released over the vast plantations of cotton, banana, and sugar cane in the region. The species present in this area do inarguably resist high levels of pollutants in the soil and hence are promising candidates for phytoremediation strategies in the region [13,30].

#### 2.1.2. El Ensayo

El Ensayo is a family-owned, rural farm located 6 km outside the city center of Chinandega, covering an area of 40 hectares. The owners of this farm are part of a group of 48 producers that supply a local cooperative with organic products, including wine, jam, honey, corn, beans, etc. The people working at El Ensayo are aware of the soil contamination problem present on their land, but since they rely on agriculture activities for their livelihood, they have no choice but to continue exploiting the agricultural fields and avoid the plots with the highest concentrations.

### 2.2. Exploration of Appropriate Phytoremediation Species for Chinandega

An inventory of autochthonous/naturally adapted plants was conducted in two polluted sites, El Picacho and El Ensayo, in Chinandega, Nicaragua. The identified plants were subsequently characterized according to their capacity to comply with a set of qualitative criteria that incorporate social, economic, and environmental aspects to facilitate the selection of appropriate species that can promote sustainable soil remediation in Chinandega through multifunctional agroforestry strategies.

#### The Selection of the Species

Thirty potential species were identified at El Ensayo (including fruit trees, firewood trees, and cultivated annual crops) and 35 species at El Picacho (mainly fruit trees and shrubs). A literature search was conducted on the identified species to assess their potential for multifunctional agroforestry/phytoremediation. Four criteria were used to determine the potential of the identified plant species; in order for a species to be considered appropriate, the literature should provide:

1. experimental data on soil phytoremediation capacity/pollution tolerance to heavy metals and other elementary pollutants,
2. experimental data on soil phytoremediation capacity/pollution tolerance to POPs and other organic pollutants,
3. data supporting that the species can be used in agroforestry systems, and
4. data supporting that the species can be used for profit (e.g., food, firewood, timber, medical purposes, or animal feed).

Of the species that were identified at Picacho and El Ensayo, the following 28 complied with the selection criteria and thus can be considered appropriate for multifunctional agroforestry/phytoremediation systems in Chinandega: *Amaranthus Spinosus* [31], *Anacardium occidentale* [32], *Arachis hypogaea* [33], *Arachis pintoi* [34], *Azadirachta indica* [35], *Brachiaria ruziziensis* [36], *Byrsonima crassifolia* [37], *Ceiba pentandra* [38], *Chrysopogon zizanioides* [38], *Cordia alliodora* [39], *Cymbopogon citratus* [40], *Erythrina poeppigiana* [41], *Gliricidia sepium* [42],

*Glycine max* [43], *Mangifera indica* [44], *Melia azedarach* [45], *Pennisetum purpureum* [46], *Persea americana* [47], *Portulaca oleracea* [48], *Psidium guajava* [48], *Ricinus communis* [48], *Saccharum officinarum* [49], *Sesamum indicum* [50], *Sida rhombifolia* [51], *Sorghum* spp. [52], *Tabebuia rosea* [53], *Tectona grandis* [54], *Tithonia diversifolia* [55], and *Zea mays* [56].

*2.3. The Four Scenarios*

The 28 species that complied with the selection criteria were combined to create agroforestry scenarios that could be economically viable and easily adapted by the farmers. In order to avoid human ingestion of toxic substances, primarily non-food species were used in the four scenarios. All scenarios assumed that no fertilizer was added.

2.3.1. Scenario 1: Shading System with Teak (*Tectona grandis*) and Patchouli (*Pogostemon cablin*)

*T. grandis* is a valuable timber tree that is frequently used in agroforestry systems. *P. cablin* needs shade and is grown for its oil that is used in perfumes and essences. It is not presently grown in Chinandega but has been successfully intercropped with *T. grandis* [57,58] and could provide a substantial income. All species-specific data of *T. grandis* are retrieved from a study where a monoculture of teak was planted on degraded soil in Costa Rica [59]. Data on yield of *P. cablin* were based on Kumar et al. [60], for the control with no fertilization. A root–shoot ratio of 0.2 was used to estimate root growth [61]. The turnover rate in the different above-ground parts was set to 0.2, and the plant is harvested two times per year.

2.3.2. Scenario 2: Alley Cropping System *Erythrina poeppigiana* and *Ricinus communis*

*E. poeppigiana* is a nitrogen-fixing tree commonly used in agroforestry systems as a shade tree or in alley cropping systems. In this scenario, *E. poeppigiana* is planted in alleys, 6 m apart, with 2 m between each tree. *E. poeppigiana* can be regularly pollarded and is valued in agroforestry systems, and thanks to its high nitrogen content in its foliage, it contributes to increased or sustained growth of the intercrop. At harvest (after a ten-year rotation), the stems would typically be used as propagation material for living fence posts where 80% of the stem is replanted and leaves and tender branches are left in the field as mulch. The carbon stored in living fence posts is not accounted for in this scenario since the sequestrations occur outside the scenario's boundaries. Species-specific data were collected from Masera et al. [62]. *R. communis* (castor bean), a species documented to function as phytoremediation for POPs [63], is used as an intercrop. The yield of *R. communis* for above-ground growth was based on Alexopoulou et al. [64], and a root–shoot ratio of 0.18 was used [61]. *R. communis* is assumed to have a life cycle of five years and a turnover rate, in the different above-ground parts of 0.2. *R. communis* remediates primarily by phytodegradation [63].

2.3.3. Scenario 3: Silvopasture System of *Cordia alliodora* and *Brachiaria ruziziensis*

Silvopasture is the agroforestry practice of combining tree plantations with grazing animals. In this scenario, *B. ruziziensis*, commonly known as Congo grass, is planted together with *C. alliodora*. *B. ruziziensis* is commonly grown as pasture in Central America, where it can be grazed on or cut for hay or fresh feed. Congo grass has good nutritional value for livestock and extensive and fibrous root systems, offering a maximum root surface area, making it ideal for phytoremediation [65].

*B. ruziziensis* has an above-ground biomass production of 6.0 Mg dry mass (DM) ha$^{-1}$ yr$^{-1}$, without the use of fertilizers [66]. For below-ground biomass, IPCC gives a root-to-shoot ratio of 1.6 for tropical grassland [67] used to recalculate the relative growth of roots. Grass species tend to phytoremediate primarily by bioaccumulation, mainly in the roots [65]. *C. alliodora* is a valuable timber tree commonly used in agroforestry, often in combination with coffee or cacao. In this scenario, the trees are planted 10 × 10 m apart not to cause too much shadow. The data on biomass growth, carbon content, and

products generated by *C. alliodora* are extracted from Masera et al. [62], with a carbon content of foliage of 0.47 Mg C per Mg DM [67]. Scenario 3 is divided into two sub-scenarios; 3 unmanaged (grass grazed by animals and is not managed in any way), and 3 managed (cut grass harvested regularly to provide feed for animals).

### 2.3.4. Scenario 4: Alley Cropping System of *Gliricidia sepium* and Amaranth (*Amaranthus* sp.)

*G. sepium* is a tree that, like *E. poeppigiana,* fixes nitrogen and can be pollarded regularly. It has many uses, such as green manure, fodder, living fence posts, and fuelwood [68]. In this scenario, the trees are planted 1 m apart in rows and 5 m between rows. The crops are planted 1 m from the tree rows to avoid adverse effects of shade and nutrient competition in rows with $0.4 \times 0.4$ m spacing between crops. *G. sepium* is pollarded with its prunings left to decompose on the spot (chop and drop mulching) as mulch. Similar to scenario 2, the stems would typically be used as propagation material for living fence posts after a five-year rotation, but the carbon sequestered by those fence posts is not included in the model. Wood density of *G. sepium* is 0.47 [17], and yield is based on Gunathilake [69], where pollarding was shown to give the highest yield, giving an average stem growth of 31.9 $m^3$ $ha^{-1}$ $yr^{-1}$ in this scenario. The root–shoot ratio of *G. sepium* is 0.49 [70]. Amaranthus is a genus comprising numerous species of herbaceous plants with edible leaves or seeds, native to Central and South America. Amaranth has shown yields of 9.7 Mg/ha with no added manure [71], resulting in a yield of 11.6 Mg/ha in the scenario when harvested two times per year and accounting for the fact that it uses 60% of the area. The turnover ratio was assumed to be 0.2 for all above-ground parts. Field experiments have demonstrated the capacity of amaranth to phytoextract a number of organic and inorganic pollutants [13,72–75].

### 2.4. Modeling of Carbon Sequestration Using CO2FIX

The carbon sequestration potential for the four scenarios was calculated using the software CO2FIX v 3.2. The software is adaptable to be used worldwide and can be flexibly used for uneven-aged, mixed-species forest-management regimes or multi-cohort systems. [62]. CO2FIX uses the dynamic soil carbon model *Yasso* which describes the decomposition and dynamics of soil carbon. The software is calibrated to represent the total stock of soil carbon without a distinction between soil layers [76]. Values used in CO2FIX were gathered from the literature using data as specific to the region, species, and climate as possible. For growth representation of the different species, carbon content, wood density, and annual growth in stem, foliage, branches, and roots were used in the biomass and soil module together with thinning and harvest intervals, climate variables such as degrees per month, average precipitation, and the month of the growing season to calculate the above- and below-ground growth of each species and scenario. Initial soil values for different soil compartments, such as non-woody litter and humus layers, were based on local conditions, and soil changes were then calculated in the models' soil module based on the below-ground inputs from the species growth and soil inputs, such as parts left to decompose after harvest. When scenarios included products, the percentage of different types of products and lifespans was inserted into the products module.

The following assumptions were made for the modeling of carbon sequestration; initial carbon in soil was set to 54 Mg C $ha^{-1}$, which is the mean soil organic carbon content in the Chinandega municipality [77], mean monthly temperature (26.1–28.7 depending on the month) and precipitation in the growing season of 1973.3 mm were inserted to reflect local conditions, and max biomass in each scenario was set to 300 Mg DM $ha^{-1}$. All scenarios assume a deforested area, and as such, no initial carbon from vegetation was inserted into the model. Soil organic carbon (SOC) content was divided to represent a similar deforested site, using initial values of 5% as non-woody litter, 0% as fine and coarse woody litter, 3% as soluble compounds, 11% as holocellulose and lignin-like compounds, respectively, 25% as humus stock 1 (simple humus), and 45% as humus stock 2 (complicated or recalcitrant

humus). A separate scenario was created for the above-ground part that is pollarded and the part (above and below ground) that is not, where root and soil biomass values of the previous rotation were used as initial values for the part that was not pollarded to more accurately represent the root biomass in CO2FIX when trees are pollarded. More information on how pollarding and coppicing are handled in the model can be found in the model description [78]. For turnover of roots, a ratio of 0.8 was used for trees and 0.9 for grass and crops, in line with Gill and Jackson [79] for tropical areas. For scenarios lacking specific data on carbon content, the IPCC 2006 Guidelines were used [67]. A summary of the cultivation regime for the used species is found in Table 1.

**Table 1.** Characteristics of the four scenarios.

| Scenario | Density of Trees per Hectare before Thinning | Thinning Interval of Trees (Year) | Rotation Period of Trees/Crops (Year) | Harvests per Year of Crop | Tree Products Included |
|---|---|---|---|---|---|
| 1: *T. grandis* and *P. cablin* | 1111 | 3, 10, 20, 30 | 40/3 | 2 | Yes |
| 2: *E. poeppigiana* and *R. communis* | 837 | Annual pruning of leaves and branches | 10/5 | 1 | No |
| 3: *C. alliodora* and *B. ruziziensis* | 100 | None | 20/1 [1] | Several [1] | Yes |
| 4: *G. sepium* and *Amaranthus* sp. | 4200 (pollarded) | Biannual pruning | 5/1 | 2 | No |

[1] Only relevant for scenario 3 managed, where the grass is harvested.

## 3. Results

The total carbon sequestered including products in the four scenarios after 100 years ranged between 68 and 219 Mg C/ha (Figure 1). The scenario that showed the highest mean carbon sequestration potential after 100 years was scenario 1, as shown in Table 2. A large fraction (46%) of the carbon sequestered in scenario 1 is from products derived when trees are harvested, used to create furniture, etc., and thus remain in the system for the lifetime of the product.

The second highest carbon sequestration scenario was scenario 4, which had larger inputs to the soil from green manure than the other scenarios due to a higher increase in SOC. If products were not accounted for, the scenarios showed more similar carbon fluctuation trends, as can be seen in Figure 1, with scenarios 1 and 3 showing larger fluctuations between years when products were included. When products were excluded, scenario 4 became more in line with scenario 1, and scenario 2 captured more carbon than scenario 3 unmanaged and managed. While scenario 3 unmanaged and managed yield showed very similar results, scenario 3 unmanaged showed a higher carbon sequestration potential than scenario 3 managed, with the difference being that the grass in the unmanaged scenario had higher biomass and input to soil than in the managed scenario where the grass was harvested regularly.

**Table 2.** Long-term mean change in carbon stock and carbon sequestration rate per scenario.

| Scenario | Soil | Total Including Products | | Total Excluding Products | |
|---|---|---|---|---|---|
| | Mg C ha$^{-1}$ yr$^{-1}$ | Mg C ha$^{-1}$ yr$^{-1}$ | Mg CO$_2$eq ha$^{-1}$ yr$^{-1}$ | Mg C ha$^{-1}$ yr$^{-1}$ | Mg CO$_2$eq ha$^{-1}$ yr$^{-1}$ |
| Scenario 1 | 0.5 | 2.2 | 8.0 | 1.2 | 4.4 |
| Scenario 2 | 0.7 | no products | no products | 0.7 | 2.7 |
| Scenario 3 unmanaged | 0.4 | 0.7 | 2.7 | 0.5 | 1.7 |
| Scenario 3 managed | 0.4 | 0.7 | 2.5 | 0.4 | 1.5 |
| Scenario 4 | 0.9 | no products | no products | 1.0 | 3.6 |

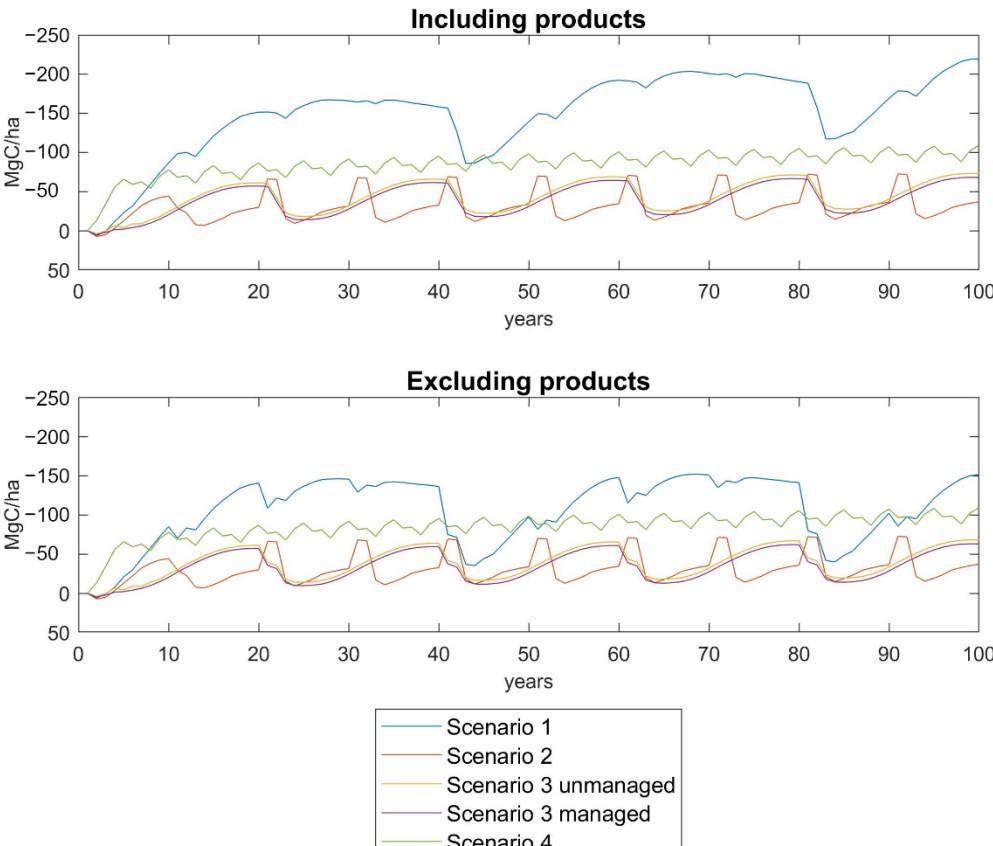

**Figure 1.** Total amount of carbon sequestered per scenario and year, including or excluding products.

The most significant carbon fluctuation was found in the biomass and soil of trees, with very little biomass in crops resulting from regular harvests. Despite this, all crops in-creased SOC but to a varying degree. The highest inputs to soil from crops after 100 years were in scenario 3 unmanaged ($33.5$ Mg C ha$^{-1}$), followed by scenario 3 managed ($29.9$ Mg C ha$^{-1}$) and scenario 4 ($12.2$ Mg C ha$^{-1}$), while crops in scenarios 1 and 2 only increased SOC by $2.7$ and $2.6$ Mg C ha$^{-1}$, respectively. In scenario 3, all the increased SOC came from the crop, vastly differing from the other scenarios where crop to soil increase varied 4–12%. This is due to the differences in how the trees influence SOC, as shown in Figure 2, where all scenarios showed declined SOC as trees were using carbon from the soil after the initial plantation. As the tree grows, SOC increased with a peak just after harvest and declined as new trees were planted. For most scenarios, the SOC increased over time from both the trees and crops, but in scenario 3, the tree slowly decreased the SOC. However, the total SOC changes in both scenario 3 unmanaged and managed were positive, in response to the higher input to the soil from the grass that mitigates the loss resulting from the trees. Of the SOC changes, the more stable ones over time were in the humus layers, with the complicated humus layer being the most stable. Fluctuations in these layers are shown in Figure 3, where scenario 4 showed a significantly higher SOC change in both the simple and complicated humus compared to the other scenarios.

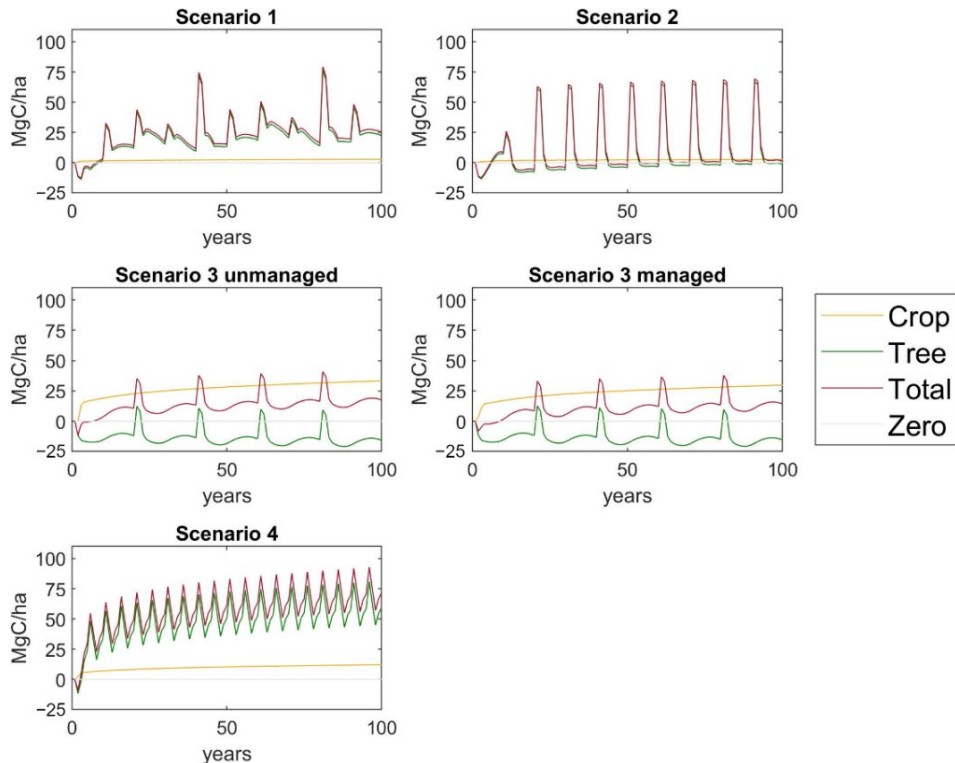

**Figure 2.** Fluctuations in the soil compartment shown as inputs from trees, crops, and total, showing how SOC varies over time in the different scenarios.

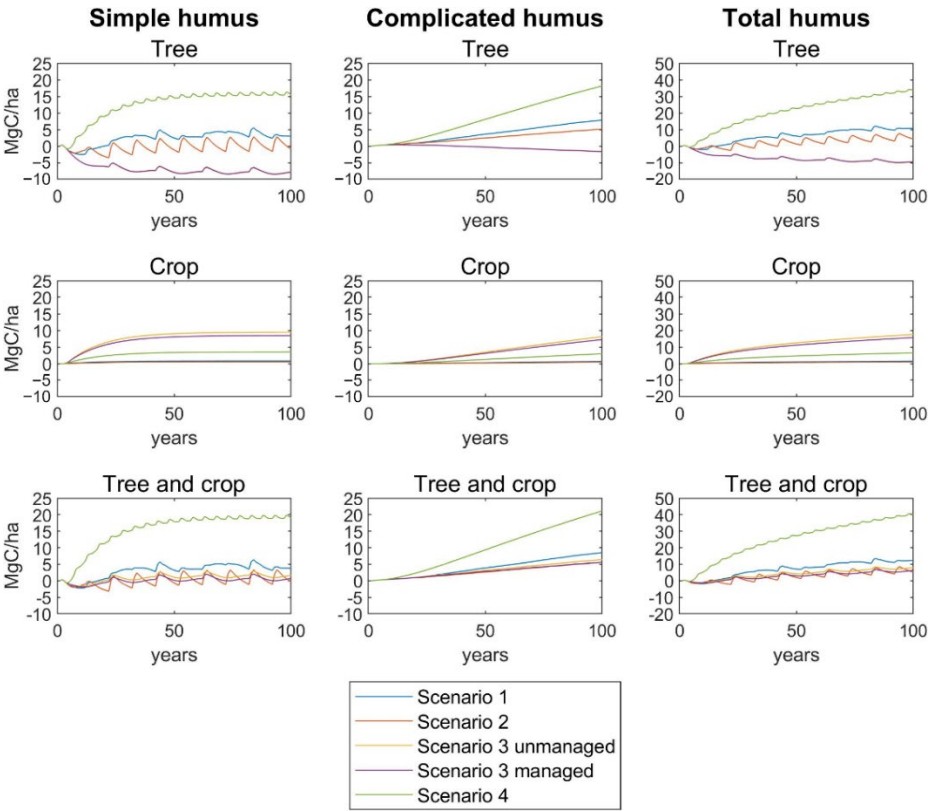

**Figure 3.** Changes in the different humus portions (simple, complicated, and total) per scenario, showing how a large portion of changes in SOC relates to tree and crop inputs to the soil.

The carbon sequestration potential varied in all scenarios over time, which can, at least partly, be explained by different thinning regimes, harvesting, and growth patterns of the different species. One example of the impact of different management practices is that perennial crops result in higher inputs to SOC compared to annual or shorter-lived crops. Another example is that higher overall carbon sequestration and SOC are given in scenario 4 compared to scenario 2, even though both trees are used as green manure, but with varying initial tree density, pruning, and rotation intervals.

## 4. Discussion

NBSs such as MAP may positively influence several socioeconomic variables and increase synergies and trade-offs [10,80,81]. The provision of reliable data to stakeholders about non-commodity outputs, such as soil remediation and carbon sequestration of multifunctional land-use systems, is crucial to support informed decisions about their implementation. This study provides data from modeling of the carbon sequestration potential of four MAP systems suggesting that carbon sequestration is a considerable non-commodity output of such systems. The quantification of carbon sequestration is insufficient to assess the complex interplay between agroecosystems and their surrounding ecosystems and social contexts. Instead, it should be considered one vital factor to be weighed against other context-specific economic, ecological, and social objectives. Before any scenario can be implemented, a thorough baseline analysis of the needs that the MAP system should address is necessary. The remediation needs should be assessed together with all other desired commodity and non-commodity outputs that the system is expected to deliver. Our proposed scenarios may be adapted for the particular needs of any location, primarily in rural areas in tropical low-income countries.

Although the carbon sequestration of an agroforestry-based phytoremediation system is typically lower than that of a tropical forest (that may sequester 1.5–5.8 Mg C ha$^{-1}$ yr$^{-1}$) [79], the proposed scenarios sequester significantly more carbon than conventional farming practices [76], in line with our hypothesis. In agriculture, crops result in small carbon fluxes as they are harvested at short intervals, which this study also shows. However, combining crops with trees significantly improves the potential of carbon sequestration in the scenarios, which has also been found in several other studies showing increased sequestration rates of 1.5–3 times compared to monoculture [21–23]. Scenario 3 can be compared with grasslands that average between 0.11–0.81 Mg $CO_2$-eq. ha$^{-1}$ yr$^{-1}$ [82], while scenario 3 shows values at least twice that, despite a very low portion of the carbon sequestration coming from the trees that have a slowly decreasing soil sequestration in this scenario. To achieve long-term carbon sequestration, the most important factor is how the crops and trees affect SOC values in general, and especially the humus layers, as they are more stable over time. With this in mind, scenario 2 and scenario 3 unmanaged show potential for long-term soil carbon sequestration. As scenarios 2 and 4 show the most increase in SOC, the use of green manure can be a good strategy to increase both long-term sequestration and soil fertility. However, if the target pollutants are elements or organic pollutants that are not phytodegraded, there is a risk that the pollutants are recirculated into the soil, offsetting the remediation. A thorough assessment of how much of the target pollutants are translocated to the green leaves and what fraction of the total extracted/degraded pollutants they constitute are thus needed. As a living fence was not part of the modeling of these scenarios, these scenarios would likely result in higher carbon sequestration rates than given, potentially amounting to around 0.3 Mg C ha$^{-1}$ yr$^{-1}$ [76] due to the carbon sequestered in the living fence post outside the scenario boundaries. The scenarios considered in this study all assume a deforested area. That means that as our scenarios show carbon sequestration rates of 0.7–2.2 Mg C ha$^{-1}$ yr$^{-1}$ (including products), they will contribute to increased carbon sequestration, as well as provision of a multitude of other benefits including economic.

Depending on the priority of goals, different scenarios are to be recommended. For faster income generation, scenarios 1, 2, and 4 provide income one to several times

a year from the production of patchouli, castor bean, and amaranth, while scenarios 1 and 3 can provide income after 40/20 years (one rotation period) from timber production. More than one intercrop could be grown together with the trees to avoid economic and ecological risks associated with growing a single crop. To improve soil health, scenarios 2 and 4, where nitrogen-fixing trees are used as green manure, can improve crop yields while simultaneously helping to provide healthy soils and increase soil organic carbon. In densely populated and highly polluted areas where little carbon can be captured due to the limited extension of the area, the main focus may be to limit health impacts from human exposure to polluted soil, making scenarios 2 and 3 (or adaptations of these) appropriate choices. In such cases where phytoremediation is the most important goal, scenario 3 managed can be a beneficial choice, although as most toxic substances may be bound to roots [65], extraction will be labor-intensive and more expensive. In sparsely populated areas with minor land-use conflicts and where the land extensions are more substantial and thus can capture more carbon, scenario 1 can be more beneficial. The choice of scenario will, as the above discussion suggests, depend on multiple factors, including the site characteristics and socioeconomic objectives.

Since MAP systems deliver several functions in addition to soil remediation and carbon sequestration, their implementation may reduce land-use conflicts in a world with declining soil resources and conflicting land-use interests. However, many technical and legal hindrances may limit large-scale implementations of MAP, including high management costs of the contaminated biomass, uncertain public acceptance, and legislation that may be an obstacle to agricultural activities on polluted soil [10,83,84]. The approach to use polluted soil for agricultural production irrefutably entails health risks that must be managed appropriately. We focused on non-edible products to avoid ingestion of toxic compounds, but dermal absorption, through skin contact or lung exposure through, e.g., inhalation of contaminated soil particles are risks that must be considered. If food crops or livestock fed to humans are included in a MAP system, caution must be taken that the pollutants are not translocated to the edible plant parts or the meat of grazing animals [10]. For all plants used for phytoremediation, the toxicity levels must be monitored to avoid exposure to humans or dissipation into the environment. Small-scale local testing can help contribute to further information on how each species contributes to the phytoremediation of POPs. MAP systems are associated with a number of potential ecotoxicological, economic, and social challenges and risks. Notwithstanding this, in many locations, it may be the most appropriate strategy to promote the increasingly urgent decontamination of polluted fields, provided that the edible products deriving from the systems do not contain harmful toxic compounds and that the non-edible biomass is taken care of without exposure to humans or dissipation into the environment.

## 5. Conclusions

Nature-based solutions, such as MAP, may produce multiple commodity and non-commodity outputs and positively influence several socioeconomic variables. Policymakers require reliable data about non-commodity outputs of land-use systems. This study sets out to provide data from modeling the carbon sequestration potential of four scenarios of MAP systems in Chinandega, Nicaragua. The total carbon sequestered including products in the four scenarios after 100 years range between 68 and 219 Mg C ha$^{-1}$ or 2.5 to 8.0 Mg CO$_2$eq ha$^{-1}$ yr$^{-1}$. The scenario that showed the highest carbon sequestration potential was scenario 1. However, all the scenarios have the potential to sequester carbon and remediate soil contamination. For income generation, scenarios 1, 2, and 4 can provide income after a few months up to a year from the crops (patchouli, castor bean, and amaranth), while scenarios 1 and 3 provide income from timber after 20 to 40 years. Overall, carbon sequestration in crops is relatively small, but the increased SOC is significant, especially in perennial crops. In scenario 3, SOC increased significantly and has the potential to remain in the soil long-term. If phytoremediation is included in multifunctional strategies that produce other commodity outputs and non-commodity outputs such as

carbon sequestration, the adoption rate is likely to increase. Agroforestry shows great potential as multifunctional systems that can be adapted to local needs to contribute to climate change mitigation, phytoremediation, and other objectives.

**Author Contributions:** Conceptualization, E.K., L.B., H.H. and A.J.; data curation, E.K. and L.B.; formal analysis, E.K. and L.B.; investigation, E.K., L.B. and G.F.-C.; methodology, E.K. and L.B.; resources, G.F.-C.; software, E.K. and L.B.; supervision, H.H. and A.J.; validation, E.K., L.B., G.F.-C. and H.H.; visualization, E.K.; writing—original draft, E.K.; writing—review and editing, E.K., L.B., G.F.-C., H.H. and A.J. All authors have read and agreed to the published version of the manuscript.

**Funding:** This research received no external funding.

**Institutional Review Board Statement:** Not applicable.

**Informed Consent Statement:** Not applicable.

**Data Availability Statement:** All input data utilized for modeling can be found with references in the materials and methods section. Output data presented in this study are available on request from the corresponding author. The outputs are also almost entirely given in figures and tables in this article.

**Acknowledgments:** We thank Martha Lacayo, Marta Jarquín Pascua, Maybis López Hernández, and the other members of the team at the Biotechnology Laboratory of UNAN-Managua, for their great support with the field research in Chinandega. We also thank Ajax Fonseca and Francisco Javier Espinoza (project leader and general coordinator, respectively) from the Chinantlan Cooperative Association for their assistance during the study visits at El Picacho and El Ensayo.

**Conflicts of Interest:** The authors declare no conflict of interest.

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
