# Peer review of "Modeling the Carbon Sequestration Potential of Multifunctional Agroforestry-Based Phytoremediation (MAP) Systems in Chinandega, Nicaragua"

_sustainability, doi:10.3390/su14094932_

Round 1

Reviewer 1 Report

Title: Modeling the Carbon Sequestration Potential of Multifunctional Agroforestry-based Phytoremediation (MAP) Systems in 3 Chinandega, Nicaragua

To

The Editor in Chief

Sustainability

Dear Sir/Madam,

I read manuscript thoroughly and submitting my comments for your kind consideration. The details of comments are as follows:
Comments

Abstract: This section is summary of the study therefore complete information to be given.

Introduction:

  • Some more good information of phytoremediation to be added in this section.
  • Hypothesis of the study to be highlighted with clear pointwise objectives.

2.1 Study site:

  • Line 91-93, I hope more information about pollutant and pesticides to be given here.

2.1.1 The Picacho airport

  • Line 100 and 101, The species present in this area, do inarguably resist high levels of pollutants in the soil and hence are promising candidates for phytoremediation strategies in the region (can you support this sentence with some citations?).
  • Line 135-141 many species suggested and can be suitable for phytoremediation system (Do you have any support of this literature from earlier studies)
  • Discussion section can be improved
  • Conclusion section can be also be improved
  • Use following citation for further inputs

Estimation of Risk to the Eco-Environment and Human Health of Using Heavy Metals in the Uttarakhand Himalaya, India Appl. Sci. 2020, 10(20),  7078; https://doi.org/10.3390/app10207078

Reviewer 2 Report

Dear authors,

Thanks so much for this paper, the topic is interested, the paper is well written and organized, however, there are some few comments to be more suitable for publication as follows: 

Abstract

Line 18 and 19: Four different agroforestry systems scenarios relevant to Chinandega were created… based on what??

Line 21: CO2eq/ha/yr please subscribe 2 and change the unit to SI units CO2eq ha-1 yr-1, and  please change the units to SI units through table 2 and the whole manuscript.

Introduction

Line 35: "the area that can be sustainably used for food production may already have been reached" is not clear

M&M

In the study site 2.1.1, The Picacho airport: what is its area?

You need to write even brief description of the environmental conditions of the selected species in each scenario

In the description of the CO2FIX model, you need to write the parameters which included in the model to get the results of each scenario.

Reviewer 3 Report

There are still overaps of discussion part in the conclusion, I suggest that the conclusion is refined more. 

The synergestic effects on biomediation seem to come off faded in the discussion and suggestions of use of biomass (in case of nitrogen fixers) as green manure could undo the biomediation. 

Use of 100 year projections shades a nice picture for the long term but not an economic one in the short and mid term as in most low income nations tree cover or a strict agroforestry systems may not be maianained for that long even if the tree component is targeted for products such as timber. Is it therefore possible on the side of authors to model more short (10, 25, 50 year) scenarios? as this would reflect the actual rotation times that farmers do practice.

Round 2

Reviewer 1 Report

Dear sir/Madam

Based on my previous review report. I found that the authors have made possible changes in the manuscript and can be considered in the Journal. I recommend the manuscript can be accepted in the Journal for publication.